# Direct observation of coherence transfer and rotational-to-vibrational energy exchange in optically centrifuged CO$_2$ super-rotors

Timothy Y. Chen [1,3], Scott A. Steinmetz [1], Brian D. Patterson[1], Ahren W. Jasper[2] & Christopher J. Kliewer [1] ✉

Optical centrifuges are laser-based molecular traps that can rotationally accelerate molecules to energies rivalling or exceeding molecular bond energies. Here we report time and frequency-resolved ultrafast coherent Raman measurements of optically centrifuged CO$_2$ at 380 Torr spun to energies beyond its bond dissociation energy of 5.5 eV ($J_{max} = 364$, $E_{rot} = 6.14$ eV, $E_{rot}/k_B = 71,200$ K). The entire rotational ladder from $J = 24$ to $J = 364$ was resolved simultaneously which enabled a more accurate measurement of the centrifugal distortion constants for CO$_2$. Remarkably, coherence transfer was directly observed, and time-resolved, during the field-free relaxation of the trap as rotational energy flowed into bending-mode vibrational excitation. Vibrationally excited CO$_2$ ($v_2 > 3$) was observed in the time-resolved spectra to populate after 3 mean collision times as a result of rotational-to-vibrational (R-V) energy transfer. Trajectory simulations show an optimal range of $J$ for R-V energy transfer. Dephasing rates for molecules rotating up to 5.5 times during one collision were quantified. Very slow decays of the vibrational hot band rotational coherences suggest that they are sustained by coherence transfer and line mixing.

Over the past few decades, there has been significant development in quantum state control of molecules and their degrees of freedom using lasers for manipulating chemical reactions and quantum information[1–5]. One method is the optical centrifuge[1], which traps and uni-directionally accelerates molecules to extreme rotational states ($E_{rot} > 3$ eV, $E_{rot}/k_B > 34,800$ K) and in some cases, dissociation[6,7]. As a consequence of being able to induce such high rotational excitation, optical centrifuges have been used to investigate these so-called "super-rotors" and their collisional dynamics and energy transfer[8–14]. In gas-phase molecular systems, energy transfer between molecules is primarily mediated by molecule-molecule collisions. For collisions involving super-rotors, the super-rotors can be thought of as molecular gyroscopes, where high angular momentum stabilizes the super-rotor orientation[13], promotes rotationally adiabatic collisions, and prolongs synchronized rotation[10]. In particular, entry into the rotationally adiabatic regime requires a molecule to rotate more than once during a collision. This violates traditional notions of molecular collisions as "sudden" and rotationally frozen[15,16]. Therefore, super-rotors are typically defined by the requirement that the molecules make at least one rotation within the duration of a collision or equivalently have an adiabaticity parameter greater than $\pi$ (see Supplementary Materials)[10]. The eventual thermalization of super-rotors can produce macroscopic phenomena such as vortex flows[12] as well as acoustic waves[11,12]. Optical centrifuges have also been used to study molecular potentials[7], remote magnetization of gases[17], control of chiral molecules[18], three-dimensional alignment of molecules[19,20], chemical reactivity[21], and new spectroscopic transitions[22].

Understanding gas phase energy transfer and the rotational structure of CO$_2$ is of primary importance due to the role of CO$_2$ as a greenhouse gas. It is also important for analyzing astronomical

[1]Sandia National Laboratories, Livermore 94550 CA, USA. [2]Argonne National Laboratory, Lemont 60439 IL, USA. [3]Present address: Applied Materials, Inc., Santa Clara 95051 CA, USA. ✉e-mail: cjkliew@sandia.gov

observations of $CO_2$ on hot exo-planets such as those recently made by the James Webb Space Telescope[23]. For space travel, shock heating of spacecraft entering extra-terrestrial atmospheres of planets such as Mars motivates the need to understand energy transfer of $CO_2$ rotational states that are populated well above 10,000 K[24]. On Earth, it was predicted that collisional inter-molecular rotational-to-vibrational (R-V) energy transfer plays a significant role in strong shock waves surrounding spacecraft during atmospheric re-entry[25–27]. Quantitative understanding of R-V energy transfer could have significant impacts on modelling of hypersonic planetary entry flows and spacecraft heat shield design. However, there has been relatively little attention paid to R-V energy transfer in comparison to vibrational-to-rotational (V-R) energy transfer[16,28]. One of the few state-to-state resolved observations of R-V gas-phase energy transfer was in vibrationally excited $Li_2$ (v = 5 and 9) vapor as a result of quasi-resonance between rotations and vibrations[29,30]. A later theoretical study showed that $Li_2$ super-rotors may induce quasi-resonant $\Delta v = 1$ transitions from the vibrational ground state[31].

It was previously reported that an ensemble of $CO_2$ super-rotors prepared by an optical centrifuge produced vibrationally excited molecules with 3 quanta in the bending mode within 50 molecular collisions[8]. Following work explored anisotropy of energy release and rotational adiabaticity in $CO_2$-$CO_2$[13] and $CO_2$-Ar/He[14] collisions. However, the question of whether the observed vibrational excitation in $CO_2$ arises from the optical centrifuge trap itself or collisional energy transfer has not been fully answered. If the latter was responsible, then it is still unknown whether rotations directly populate vibrational states and on what time scale this occurs due to the limited temporal resolution of previous experiments. Measurements of centrifuged $N_2O$ have shown that 1 quanta of bending vibrations in the $J = 68$ state was populated after 6 collisions[22]. However, this vibrational state is populated at ambient temperature, so it is unclear whether this was the result of R-V energy transfer or pure rotational energy transfer. Furthermore, the full rotational distribution of the centrifuged molecules was not measured in these studies due to the limited scanning range of the infrared diode laser. Therefore, it is still unknown what ultimate rotational frequency of optically centrifuged $CO_2$ can reach and whether known spectroscopic constants of $CO_2$ hold at super-rotor rotational frequencies.

In this work, we utilize ultrafast pure rotational coherent anti-Stokes Raman scattering (CARS) to monitor the entire rotational coherence established by the optical centrifuge pulse with tens of picoseconds (ps) of temporal resolution[10,32–34]. We use both the time-domain coherence decays and the CARS frequency-domain spectra to probe the molecular energy transfer dynamics. We simultaneously observe the entire rotational energy ladder of $CO_2$ from $J = 24$ to $J = 364$ with rotational energies up to 6.14 eV (-49,500 $cm^{-1}$, $E_{rot}/k_B = 71,200$ K). We present state-resolved observations of $CO_2$ spinning with energies beyond the ground-state bond dissociation energy as well as coherence decay measurements of molecules rotating up to 5.5 times during a single collision. This is more than three times as many rotations during a single collision as compared to past studies[10,35]. Using high frequency resolution nanosecond (ns) probing and detection from a highly-dispersive virtually imaged phased array (VIPA) spectrometer[36], the centrifugal distortion constants ($D$, $H$, and $L$) for $CO_2$ are revised. We use this data to resolve an ongoing discrepancy in the literature concerning the ground state spectroscopic constants determined from high-resolution ro-vibrational infrared measurements[37–39]. Time-resolved coherence decay measurements were conducted with a 65 ps probe laser. After -500 ps or 3 mean collision times at 380 Torr, odd rotational energy level peaks corresponding to vibrationally excited $CO_2$ emerge in the pure rotational CARS spectra. We use the VIPA spectrometer to analyze the hot band structure, limited only by the spectral width of the probe pulse (0.3 $cm^{-1}$). We find that 3–6 vibrational quanta in the $v_2$ bending mode are necessary to describe the

width and positions of these peaks. Thus, we directly observe that molecular super-rotors can transfer significant amounts of rotational energy to vibrational energy within a few collisions. Quasi-classical trajectory simulations[40–42] show that moderately excited $CO_2$ super-rotors are the primary drivers of R-V energy transfer with average per-collision probabilities of R-V energy transfer of up to 25%.

## Results

### Driving $CO_2$ to extreme angular momentum

An example spectrum taken in 380 Torr of $CO_2$ at a probe delay of 100 ps is shown in Fig. 1a. A wide distribution of rotational states were excited by the optical centrifuge. At the tail end of the distribution, rotational states up to $J = 364$ were detected. The presence of a centrifugal barrier prevents the $CO_2$ molecules from dissociating at these rotational energies like in $Cl_2$ and other molecules[6,7].

The observed Raman shifts can be derived from following expression for the rotational energy and the S-branch selection rule of $\Delta J = 2$:

$$E_{rot}(J) = B_v J(J+1) - D_v(J(J+1))^2 + H_v(J(J+1))^3 + L_v(J(J+1))^4 \quad (1)$$

where $B_v$, $D_v$, $H_v$, and $L_v$ are the vibrational level dependent rotational constant and centrifugal distortion constants, respectively. Note that sensitivity to the higher order centrifugal constants $H_v$ and $L_v$ requires high spectral resolution or large J. Details on the spectral fitting can be found in the Supplementary Materials. For reference, the CDSD-1000 database used transitions up to $J = 145$ in their fitting algorithm[43] and transitions up to $J = 100$ were measured in past $CO_2$ optical centrifuge work[13]. This motivates the question of whether spectroscopic constants determined at lower J can extend far into the super-rotor regime.

In Fig. 1b, the predicted line positions of $CO_2$ using a centrifugal distortion corrected rotational constant[44] were plotted alongside the experimental spectrum. Using a grating spectrometer, the existing centrifugal coefficients were accurate to within the spectrometer resolution (0.4 $cm^{-1}$/pixel). Errors in the Raman shift from a rigid rotor assumption were as large as 61.5 $cm^{-1}$ ($\Delta E_{rot} = 0.29$ eV, see the Supplementary Materials Fig. S4) which was more than twice that previously found for optically centrifuged $N_2$ and $O_2$[9]. This demonstrates that the fastest spinning $CO_2$ molecules observed here are highly distorted by centrifugal forces.

To obtain higher spectral resolution, a recently developed high-resolution VIPA spectrometer for CARS[36] was used. The VIPA spectrometer was equipped with with a 0.5 $cm^{-1}$ free spectral range (FSR) VIPA and had a 0.0025 $cm^{-1}$/pixel resolution and a seeded nanosecond Nd:YAG laser was used for the probe (0.003 $cm^{-1}$ linewidth). Every peak between $J = 26–364$ was well-resolved (see the Supplementary Materials, Fig. S5). With the advent of atomic clock-stabilized frequency combs, highly accurate and SI-traceable infrared transition frequencies of $CO_2$ are now available in the literature[37–39]. However, the rotational and centrifugal distortion constants of ground state $CO_2$ were determined through multi-vibrational band fits of ro-vibrational transitions involving $J < 90$. This has led to a discrepancy in the ground state rotational spectroscopic constants thus far (see the Supplementary Materials, Table S1). The ability to directly measure the rotational Raman shifts up to $J = 364$ with high resolution presents an opportunity to resolve this discrepancy.

From Fig. 1c, using existing rotational and centrifugal constants[37–39,44] resulted in a large spread in Raman shift residuals. Some errors were as large as 0.59 $cm^{-1}$ at $J = 364$. Only the constants fitted in[38] reproduced our data well, but there was still up to 0.011 $cm^{-1}$ error at high J. Since rotational states up to $J = 364$ were not considered previously, we fit a new set of first to third order centrifugal constants to our data ($D$, $H$, and $L$) with the value of $B$ taken from[37]. Using the fitted centrifugal distortion constants eliminates the observed error

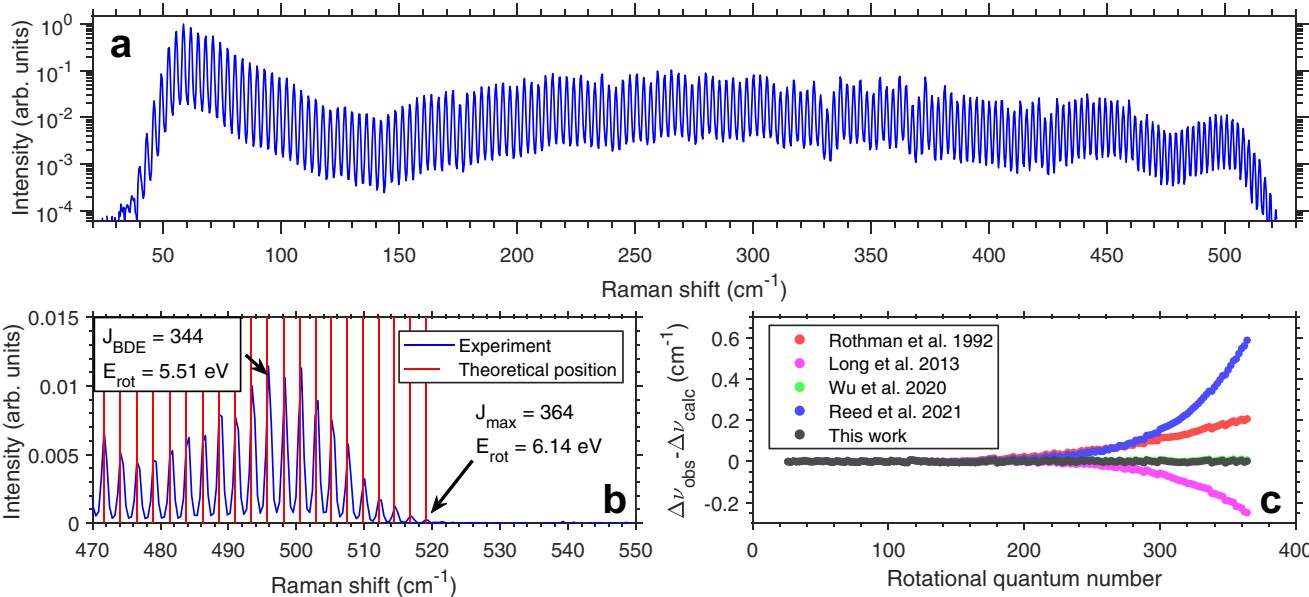

**Fig. 1 | Centrifuged $CO_2$ rotational spectrum and comparison of fitted centrifugal constants with the literature. a** Example rotational CARS spectrum of nascent centrifuged $CO_2$ at a 100 ps probe delay and pressure of 380 Torr measured using a grating spectrometer. A log scale is used to show the entire rotational distribution. States with $J < 24$ are not visible due to attenuation by the edge of the short-pass filter. **b** Close-up view of the high energy tail and comparison of calculated spectral positions and experiment. The rotational energy level with the same energy as the bond dissociation energy is denoted as $J_{BDE}$. **c** Difference between measured and predicted Raman shifts of centrifuged $CO_2$ at 50 Torr for literature spectroscopic constants and with newly fitted centrifugal distortion constants. A high resolution VIPA spectrometer was used with -0.0025 cm⁻¹/pixel resolution along with a seeded ns Nd:YAG probe laser. The constants of Wu et al. closely match those of the current work and produce an error of 0.011 cm⁻¹ at high J. Constants determined in this work produce an error of 0.0065 cm⁻1 at high J (limited by the VIPA resolution).

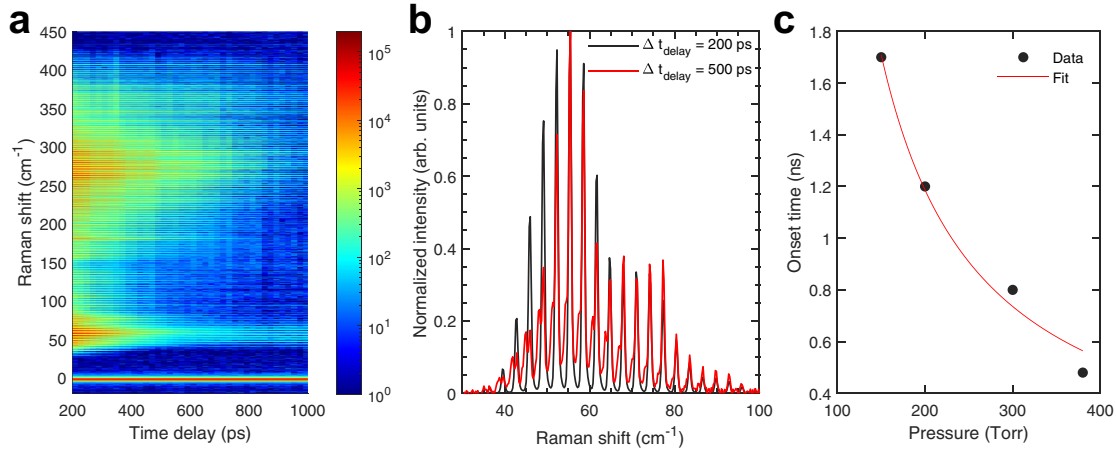

**Fig. 2 | Probe delay scans show that onset of vibrationally excited $CO_2$ depends on pressure. a** Probe delay scan of centrifuged $CO_2$ coherence at 380 Torr. **b** Appearance of vibrationally excited $CO_2$ hot bands at a 500 ps probe delay for centrifuged $CO_2$ at 380 Torr. **c** Onset times for the hot band appearance as a function of pressure and fit by a 1/P function.

for all observed rotational states ($J_{max} = 364$) to within 0.01 cm⁻¹ and within 0.0065 cm⁻¹ for the highest rotational states.

The values of our fitted constants and their comparisons with those found in the literature are summarized in the Supplementary Materials (Table S1). Only the constants determined by Wu et al.[38] agreed well with our measured Raman shifts and fitted constants. The constants determined by Long et al.[37] do not include $L$, which explains the deviation at high $J$. Unexpectedly, the spectroscopic constants with the lowest reported uncertainty[39] had the highest Raman shift error at high $J$. The $H$ and $L$ constants determined by Reed et al.[39] are significantly larger than those determined by our fits and by Wu et al.[38] Agreement with our data is uniquely sensitive to the values of $H$ and $L$ as a result of having resolved over three times more rotational states

than in previous studies. The Cs-stabilized frequency comb measurements[37,39] are presumed to be more accurate than stabilized by Rb[38] due to Cs being a SI primary frequency standard. Therefore, we attribute this error at high $J$ to the differences in the multi-band fitting routines and data inputs which led to divergent solutions, particularly in $H$ and $L$.

### Rotation-to-vibration collisional energy transfer

Shown in Fig. 2a is a map of the rotational CARS spectrum as a function of probe delay from the centrifuge pulse using a 65 ps pulse width probe laser. At high Raman shifts, it is apparent that super-rotors dephase slower than slower rotating molecules that fall out of the molecular trap early on. However, there is a response from molecules

at Raman shifts between 40 and 100 cm$^{-1}$ that extends up to a probe delay of 1 ns. A close up evaluation of the spectra at 200 and 500 ps is shown in Fig. 2b. At a probe delay of 500 ps, side bands appear in the spectrum that are not present in the 200 ps spectrum. The rotational CARS spectrum of $CO_2$ only contains even rotational energy levels at room temperature and requires high temperatures to populate bending vibrational modes for odd rotational energy levels to appear[45]. With no vibrational excitation, odd rotational energy levels lie in between the even rotational energy levels with a spacing of 1.5 cm$^{-1}$. However, additional vibrational quanta modifies the rotational constant through the molecule's moment of inertia[44]. Here, the observed shifts are ~1.1 cm$^{-1}$, which suggest significant vibrational excitation in these so-called "hot bands."

By a probe delay of 200 ps, the rotational coherences evolved in a field-free environment. The appearance of hot bands after 500 ps suggests that they are the result of collisional rotation-to-vibration energy transfer. To investigate this, the pressure was varied and the onset times of the hot bands in the rotational spectrum are shown in Fig. 2c (see the Supplementary Materials for onset time determination). As pressure increased, the time it took for the hot bands to appear decreased according to an inverse proportionality with the pressure. This is consistent with a collisional energy transfer description, where increasing pressure decreases the mean free path and the time between collisions.

### High resolution detection using VIPA-resolved CARS

To maximize the amount of information from the rotational spectra, the VIPA spectrometer was used to disperse the signal as shown in Fig. 3. Since the hot band features are within 1.5 cm$^{-1}$ of the ground state peaks, a VIPA with a free spectral range of 2 cm$^{-1}$ (dispersion of 0.01 cm$^{-1}$/pixel in the vertical direction) was used to resolve their structure. As a result, these spectra are probe pulse bandwidth limited (0.3 cm$^{-1}$). At a probe delay of 100 ps, only even ground state rotational lines are visible and the influence of centrifugal distortions can be seen in the fine dispersion axis as an upward curve at high $J$. At a 500 ps probe delay (Fig. 3b), the vibrational hot bands are clearly visible

alongside the ground state rotational lines. Due to the 2-D dispersion, the hot band peaks are well isolated, and the hot bands grow into the spectrum at a 450 ps probe delay (see the Supplementary Materials, Fig. S6).

One may wonder if high rotational states ($J > 200$) are a prerequisite for observing the hot bands. On one hand, energetic rotational states provide a reservoir of energy necessary for vibrational excitation. On the other hand, energy gap laws dictate that energy transfer with large energy gaps are more improbable than those with smaller ones. Therefore, the fastest super-rotors may not play much of a role here. In Fig. 3c, the centrifuge was cut by more than half and could only spin molecules to $J = 114$. Under such conditions, the hot bands still appear in the rotational spectrum. Furthermore, at a probe delay of 1 ns (shown in Fig. 3d), the even rotational states have completely lost coherence and cannot be detected. Only the vibrational hot bands remain in the spectrum. As will be discussed later, these hot bands are unexpectedly long-lived. Finally, a transform-limited fs "kick" (shown in Fig. 3e) was used as a baseline comparison. No vibrational hot bands arise at a probe delay of 450 ps.

To analyze the structure of the hot bands, the VIPA images were "unwrapped" from 2-D dispersed images into a high resolution spectrum as shown in Fig. 4a. A probe delay of 700 ps was chosen to minimize contamination of the hot band structure from even rotational lines while retaining some signal at these lines. A striking feature of the unwrapped spectrum is that the hot band is spectrally broad and spans more than 0.8 cm$^{-1}$. This is much larger than the probe bandwidth (0.3 cm$^{-1}$). This suggests that the hot band is composed of a distribution of vibrational states. Also shown are possible vibrational state assignments to the observed spectrum using known rotational constants of $CO_2$[44]. Surprisingly, 3−6 vibrational quanta in the bending modes are required to describe the hot band distribution. Lower energy states likely play a role in the lower frequency wing, but the peak is well-described by large excitations in the bending mode. This suggests that excitations of bending modes are preferred. This could proceed from in-plane collisions where imparted angular momentum can be taken up by a bending vibration rather than additional angular

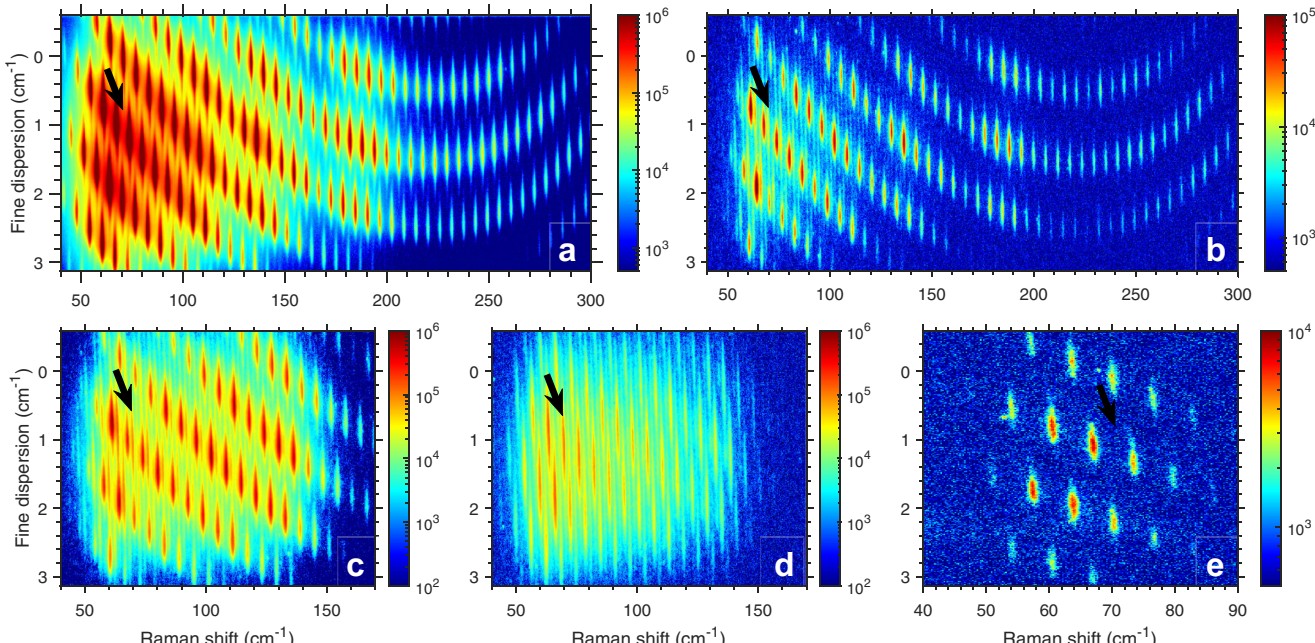

**Fig. 3 | VIPA spectra of centrifuged $CO_2$. a** VIPA spectrum of centrifuged $CO_2$ using the full optical centrifuge pulse bandwidth at a 100 ps probe delay ($J_{max} > 200$). **b** Same as (**a**) but with a probe delay of 500 ps. **c** VIPA spectrum of centrifuged $CO_2$ using a frequency-cut optical centrifuge pulse at a 500 ps probe delay ($J_{max} = 114$). **d** Same as (**c**) but with a 1 ns probe delay. **e** VIPA spectrum of $CO_2$ coherently excited by a 50 fs pulse "kick" at a probe delay of 450 ps. Hot bands do not develop for such a pulse. Arrows pointing at the $J = 43$ hot band location are shown in all sub-plots for reference.

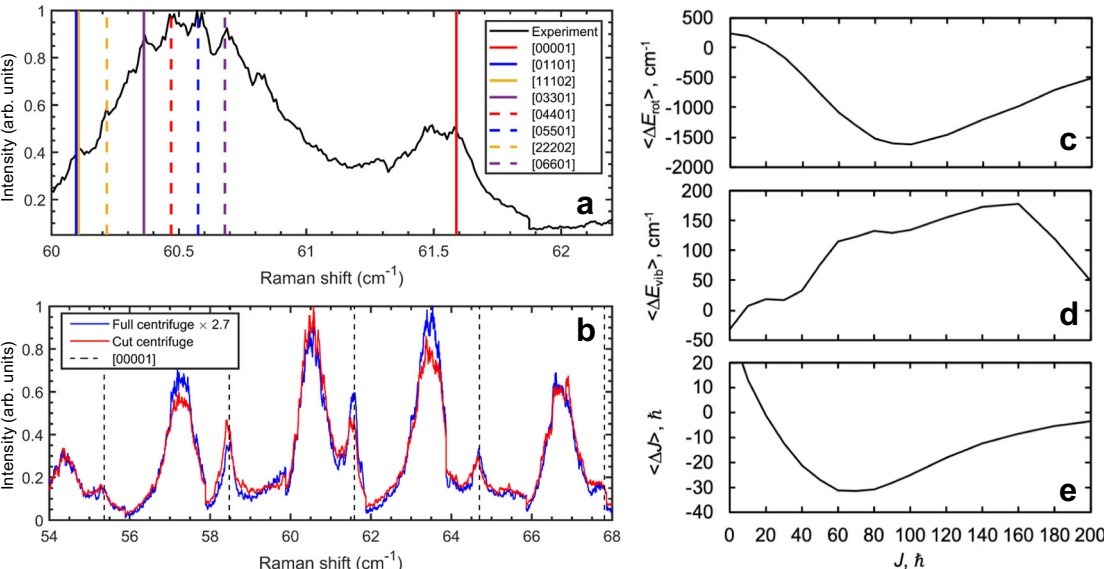

**Fig. 4 | Unwrapped VIPA spectra and predicted collisional energy transfer of different rotational states of $CO_2$. a** Unwrapped VIPA spectrum of the $J = 38$ transition of $CO_2$ at a probe delay of 700 ps. The ground state $J = 38$ line and selected vibrational states of $J = 37$ are shown for reference. The legend follows Air Force Geophysics Laboratory (AFGL) notation ($[v_1, v_2, l, v_3, r]$), and the vibrational states are ordered from least to most energetic. **b** Comparison of pure rotational spectra between full and frequency-cut optical centrifuge pulses. Positions of ground state even rotational transitions are shown for reference. Predicted collisional (**c**) rotational energy, (**d**) vibrational energy, and (**e**) angular momentum transfer with a 300 K thermal bath of CO2. The *x*-axis represents the initial state of the target molecule, and its collision partner was randomly sampled from the thermal bath.

velocity. The ratios between the hot bands and ground state peaks are also not dramatically affected by whether the centrifuge pulse is cut short to $J_{max} = 114$ or not (see Fig. 4b). This is consistent with the energy gap law argument that predicts the fastest spinning super-rotors do not significantly participate in the inelastic energy transfer dynamics. Further frequency cutting of the centrifuge pulse ($J < 80$, $\Delta\nu_{Raman} < 126$ cm$^{-1}$) results in hot bands shifting to lower rotational quantum numbers ($J < 50$, $\Delta\nu_{Raman} < 80$ cm$^{-1}$, see the Supplementary Materials, Fig. S7).

**Quasi-classical trajectory simulations**

To provide additional insight into the collisional energy transfer, quasi-classical trajectory simulations were performed for $CO_2$-$CO_2$ collisions. As shown in Fig. 4c, the average rotational energy transfer of $CO_2$ in a 300 K thermal $CO_2$ bath has a maximum for $CO_2$ molecules with initial rotational quantum numbers around $J = 90$. Correspondingly in Fig. 4d, the average R-V energy transfer also reaches a maximum and plateaus in this region. As shown in Fig. 4e, changes in rotational quantum number up to $\Delta J = -30$ are possible. This agrees with recent 2-D IR measurements of $CO_2$[46], where $\Delta J > 20$ were shown to occur within a single collision. Figure 4d can be interpreted as a probability of rotation-to-vibration energy transfer in a single collision (see Supplementary Materials), with 7% probability and 25% probability for one quantum of bending excitation with initial $J = 40$ and $J = 100$, respectively. The simulations show that moderate rotational excitation is efficient in driving R-V energy transfer. Furthermore, the highest rotational states ($J > 200$) do not contribute much to R-V energy transfer. This explains why experimentally cutting the centrifuge pulse to $J_{max} = 114$ does not eliminate the hot bands as these rotational states still have high probability of R-V energy transfer. The fact that $\Delta J = -30$ at $J = 80$ also clarifies why further cuts to $J_{max} = 80$ limits the maximum $J$ of the hot band lines to <50. On average, molecules with initial rotational states from $J = 60$ to $J = 80$ will lose 30 quanta of rotational energy and may simultaneously gain vibrational energy. This makes observations of hot bands at $J > 50$ unlikely when $J_{max} = 80$.

The rotational transition linewidths extracted from exponential fits to the coherence decays are shown in Fig. 5a. The measured linewidths at low $J$ agree well with those measured by ps CARS coherence decays[34] as well as those from high-resolution infrared absorption spectroscopy[44]. However, the fits derived from infrared measurements only extend to $J = 121$. Here, we have extended the available linewidth data by nearly three times to $J = 362$, with $CO_2$ molecules transitioning into super-rotors at $J = 66$ as shown in Fig. 5b. For reference, the linewidths measured in a 1900 K furnace reached a maximum $J = 86$[35]. As a consequence, the measured linewidths involve molecules spinning more than five times during a single collision, far above the super-rotor threshold of one rotation during a collision. Previous frequency-resolved linewidth measurements using optical centrifuges reached at most 1.8 rotations during a collision[10]. The super-rotors dephase an order of magnitude slower than slower rotors and the shape of the overall distribution appears to be exponential.

While super-rotor dephasing rates have previously been found to dephase slower[10,47], the hot bands also dephased more than order of magnitude slower than those in the ground state. We were able to observe these hot bands persist even at a 1.6 ns probe delay after the centrifuge pulse(see the Supplementary Materials, Fig. S8). The vibrational state specific $CO_2$ rotational linewidths do not change by more than 15%[48]. Any vibrationally excited $CO_2$ should then be visible in the spectrum at early time delays and decay together with the vibrational ground state rotational lines. However, we do not observe any hot bands at early time delays and instead see vibrationally excited $CO_2$ grow into the spectrum after several collisions occurred (see Supplementary Materials Fig. S6). This removes the possibility that the optical centrifuge pulse initially populated these vibrationally excited states and that our observations are the result of differences in coherence decay times.

From Figs. 2–4, it was established that the hot bands are the result of collisional R-V energy transfer. However, we must also consider that the observed molecules need to be *coherently* populated in order to be observable by coherent Raman scattering, i.e., spinning in phase with the initially excited rotational wavepacket. Figure 3e shows that random collisions cannot appreciably populate the observed hot bands.

Once all of the ground state coherences have dephased, the only coherent molecules left are those in the hot bands. Collisions with

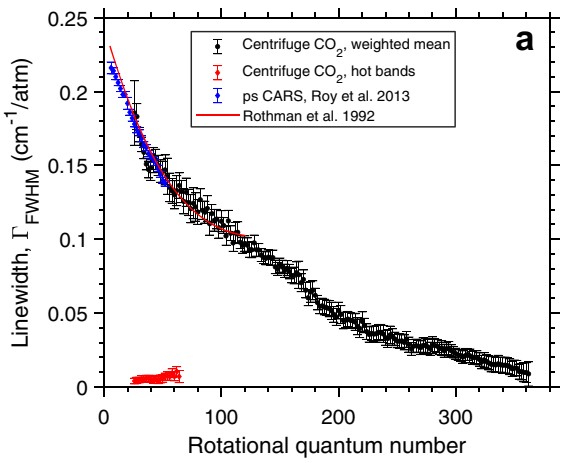
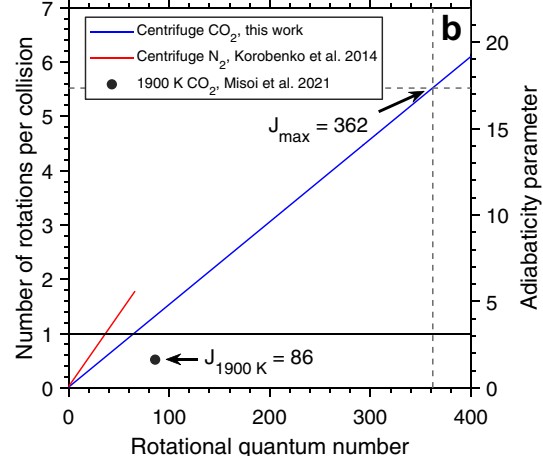

**Fig. 5 | $CO_2$ linewidths measured up to $J = 362$.** Characteristic linewidths of the coherence decays of even rotational transitions and the hot bands (**a**). Number of rotations during a single collision and corresponding adiabaticity factors for measured transition linewidths (**b**). Data from the literature are plotted for reference[34,35,44] and error bars correspond to uncertainties in the decay fits. The expression of Rothman et al. extends up to $J = 121$. Dashed lines in (**b**) correspond to the maximum rotational quantum number for the linewidths measured in this work. A horizontal line corresponding to the 1 rotation per collision threshold (adiabaticity parameter of $\pi$) is plotted for reference as done in ref. 10.

vibrational ground state molecules should dominate the coherence or energy transfer. With all ground state coherences decayed away, how can the hot bands still persist in the spectrum? To explain their unusually long coherence life times, we note that so-called "motional narrowing" of $CO_2$ Q-branch spectra into a single peak close to the Doppler limit and theory of this phenomenon has been reported in the literature[49–51]. Time-domain measurements of the $CO_2$ Q-branch showed that this narrowing manifests itself as the smoothing of quantum beats and extension of the coherence life time via coherence transfer between rotational transitions in the Q-branch[52]. This requires a highly congested spectrum where the spectral width of each line overlaps with its neighbors and the lines mix. Here, we have an analogue of a congested Q-branch where the vibrationally shifted S-branch transitions can be as close as 0.01 cm⁻¹ (see the Supplementary Materials, Fig. S10). This is much narrower than the linewidths measured in Fig. 5.

This interpretation of the data requires that molecules within these hot bands undergo vibrational energy transfer with bath gas molecules without loss of rotational coherence or rotational coherence transfer with vibrationally excited bath gas molecules. Dephased rotationally excited molecules still exist as part of the bath gas and are available to either provide or take away vibrational energy. Therefore, the complete dephasing of ground state $CO_2$ does not necessarily impede the persistence of the hot band coherences. Full validation of this mechanism will require measurements of the bath gas such as with ultrafast infrared absorption or an additional femtosecond pump-Stokes pulse to re-align the dephased molecules.

In summary, R-V energy exchange and coherence transfer in optically centrifuged $CO_2$ was investigated using high spectral and temporal resolution coherent Raman scattering and quasi-classical trajectory simulations. From the VIPA-resolved measurements, a high-accuracy set of centrifugal distortion constants was determined with <0.0065 cm⁻¹ of error at the highest rotational states. This resolved the disagreement between constants inferred from multi-vibrational band fits of atomic clock-stabilized frequency comb measurements in the literature. Furthermore, using ultrafast coherent Raman scattering, we have observed vibrationally excited $CO_2$ after three mean collision times as a result of R-V energy transfer. Trajectory simulations showed that R-V energy transfer probability peaked at moderate rotational excitation, with up to 25% probability per collision. The fastest spinning $CO_2$ molecules ($J > 200$) were found to not have a significant role in R-V energy transfer. Lastly, the hot band transitions were found to

persist unusually long, and these transitions were hypothesized to be sustained by coherence transfer. The presented measurements and simulations open the door to fully resolved pure rotational molecular structure and centrifugal distortions beyond bond-dissociation energies as well as studies of super-rotor R-V energy transfer and coherence transfer occurring on the time scale of molecular collisions.

## Methods
### Experimental methods
The optical centrifuge pulse was generated from the output of a chirped pulse Ti:Sapphire regenerative amplifier (Coherent, Legend Elite), similar to previous implementations[6–9]. The schematic diagrams can be found in Figs. S1–S3. The centrifuge pulse was centered around 790 nm and had a transform-limited duration of 50 fs. Here, we used a 0.8 mJ portion of the uncompressed chirped pulse and sent it into a 7-pass power amplifier. A 25 mm diameter Brewster-cut Ti:Sapphire crystal was end-pumped by two 90 mJ relay-imaged 532 nm pulses created from a 10 Hz ns Nd:YAG laser (Surelite II, Continuum) and a 50/50 high energy plate beam splitter. The 532 nm pulses were time delayed by 13 ns from one another to prevent damage to the crystal. The resulting pulse was amplified to 10 mJ. For the measurements of Figs. 1 and 5 as well as Figs. S4, S5, and S9, the seed and pump energies were increased to 1 mJ and 120 mJ, which resulted in an 18 mJ amplified pulse. The final amplified pulse was sent into a beam expander and the 4-f pulse shaper. One leg maintained the positive chirp, while the other reversed the chirp. A business card mounted on a translation stage was used to terminate the centrifuge pulse early by blocking part of one leg at the Fourier plane as in ref. 9. The two pulses were recombined in a polarizing beam splitter cube with orthogonal linear polarizations. The overall transmission efficiency was approximately 40%. The combined pulse had an energy of either 4 or 7 mJ and was passed through a $\lambda/4$ waveplate to circularly polarize the light. The resultant optical centrifuge pulse was then focused into a gas cell filled with CO2 at pressures ranging from 50 Torr to 380 Torr. The purity of the CO2 gas cylinder was 99.99% (Coleman Instrument grade, Matheson). The CARS spectra were generated by a 65 ps pulse from a ps Nd:YAG regenerative amplifier and was collinearly phase-matched with the centrifuge pulse. Probe energy up to 1 mJ was used. For high resolution spectra, a seeded Nd:YAG (Powerlite, Continuum) was used which provided 6 ns pulses with energies of 10 mJ. The linewidth of the seeded Nd:YAG laser was specified by the manufacturer to be 0.003 cm⁻¹. Both probes had a horizontal linear polarization.

A dichroic mirror was used to separate the probe and CARS signal from the centrifuge pulse. The oscillators of the ps Nd:YAG and Ti:Sapphire regenerative amplifiers were phase locked to within 1 ps, and the relative probe pulse delays were controlled electronically (PDL-100A, Colby Instruments). The seeded Nd:YAG was triggered by a delay generator (DG645, Stanford Research Systems) synchronized to the Ti:Sapphire regenerative amplifier. The CARS signal was measured by a Czerny Turner spectrometer (Horiba, iHR550) and charge coupled device (CCD) camera (Newton 940, Andor). For all measurements presented here, 500–4000 laser shots were averaged per spectrum. The linewidths reported in Fig. 5 are the average of 6 delay scans and the means of the fitting uncertainties were used as the error bars. Two short-pass filters (SP01-561RU, Semrock) were used to reduce the probe intensity to the same order of magnitude as the CARS signal.

For high spectral resolution images, a virtually imaged phased array (VIPA) spectrometer was used to disperse the CARS signal[36]. The CARS signal was spatially filtered with a 200 micron pinhole and imaged into the VIPA (LightMachinery) using an achromatic lens and a cylindrical lens with focal lengths of 100 mm and 300 mm, respectively. The VIPA dispersed signal was then focused into a Czerny-Turner grating spectrometer (Shamrock 500i, Andor) with two orthogonally oriented cylindrical lenses. The focal lengths of theses lenses were 75 mm and 500 mm. The 500 mm lens imaged the VIPA dispersion onto the slit in the vertical direction, while the 75 mm lens focused the light onto the slit in the horizontal direction. The spectrometer slit length was oriented vertically and the grating dispersion direction was horizontal. The cross-dispersed CARS signal was detected by an intensified CCD camera (iStar, Andor) with a gate width of 100 ns. For nanosecond probing, a VIPA with a 0.5 cm$^{-1}$ free-spectral range was used, while for picosecond probing, a 2.0 cm$^{-1}$ free spectral range was used. The VIPA free spectral ranges were specified by the manufacturer to be $0.5 \pm 0.005$ and $2 \pm 0.02$ cm$^{-1}$ and were measured to be 0.4965 cm$^{-1}$ and 1.9954 cm$^{-1}$ using CO and N2 rotational Raman spectrum as calibrations, respectively. Rotational constants of CO have been fit using high frequency resolution data with uncertainties ranging from $10^{-3}$ cm$^{-1}$ to $10^{-8}$ cm$^{-1}$ and rotational states up to J = 132[53]. For reference, we observed optically centrifuged CO up to J = 67, which spanned across the entire CCD chip. Therefore, CO was used as the calibration gas for the 0.5 cm$^{-1}$ FSR measurements to ensure the highest level of accuracy for determining new centrifugal distortion constants. The entire rotational expansion (up to $O_0 J^7(J+1)^7$) for vibrational ground state $^{12}C^{16}O$ from[53] was used in the calculation of rotational energies. For the 2 cm$^{-1}$ FSR VIPA, N2 was sufficiently accurate for frequency calibration, particularly in the low J regions where vibrational hot bands were observed. The N2 rotational constants used were $B = 1.99824$ cm$^{-1}$ and $D = 5.7610 \times 10^{-6}$ cm$^{-1}$ [54]. For observations of the highest rotational states in this work, 1800 gr/mm gratings were used in order to detect the entire spectrum on the CCD chip. Otherwise, the spectra were dispersed with a 2400 gr/mm grating to maximize the spectral resolution. Frequency axes were determined by a second order polynomial fit to the observed rotational Raman line positions for each calibration gas.

Linewidths were obtained by fitting an exponential function to each coherence decay. The $1/e$ time constant can be related to the Lorentzian linewidth using the following relation:

$$\Gamma = 1/2\pi c\tau \qquad (2)$$

where $\Gamma$ is the Lorentzian full-width-at-half-maximum linewidth, $c$ is the speed of light, and $\tau$ is the $1/e$ time constant of the coherence decay.

### Numerical methods

Collisional energy and angular momentum transfer was theoretically calculated as follows. The required full-dimensional $CO_2 + CO_2$ potential energy surface was generated using strategies described and validated previously[42,55]. The interaction potential was represented using permutationally invariant polynomial (PIP) expansions[56] and the computer code PIPPy[57]. The training data set consisted of 72,000 counterpose corrected SAC MP2/CBS energies, where the complete basis set (CBS) limit was estimated from aug-cc-pVDZ and aug-cc-pVTZ energies and a scaling all correlation (SAC,[40]) correction factor of 1.08 was employed to improve the binding energy. At this level of theory, the $CO_2$ dimer binding energy is 513 cm$^{-1}$, in good agreement with a recent CCSD(T)/CBS calculation (512 cm$^{-1}$,[56]). Convergence with respect to the order of the PIP expansion was demonstrated by comparing results for 3rd and 4th order expansions; the 4th order expansion is four times larger than the 3rd order one, and both expansions gave collisional outcomes that agreed within the present statistical uncertainties; attention is limited to results for the 4th order expansion below. The fitted $CO_2$ monomer potential was adjusted to give harmonic frequencies ($\omega$ = 668, 1282, and 2455 cm$^{-1}$) and an equilibrium rotational constant ($B_e = 0.390$ cm$^{-1}$) in good agreement with experiment ($\omega$ = 672, 1351, and 2396, and $B_e = 0.390$ cm$^{-1}$,[58]).

Quasi-classical trajectories (QCTs,[59,60]) and the dynamics code DiNT[42] were used to prepare state-specific initial conditions with zero-point energy in each vibrational mode. We confirmed that the highly averaged collision outcomes of interest here were similar when vibrational states were instead selected from a thermal distribution at 300 K. One of the pair of $CO_2$ colliders (the target) was prepared with a well-defined initial rotational state $J$, while the initial rotational state of the other $CO_2$ collider (the bath) was sampled from either a thermal distribution at 300 K ($J < 50$) or from the experimentally determined high-$J$ distribution ($J \approx 60$–360). Numerical parameters controlling integration and the initial and final collider separation were chosen such that the total energy was conserved to better than 2 cm$^{-1}$, well below the finite-ensemble statistical error of 10 cm$^{-1}$ (7200 trajectories). The final total energy and angular momentum $J'$ of the target $CO_2$ were computed, and the change in the total energy $\Delta E$, the change in the rotational energy $\Delta E_r = B0(J'(J'+1) - J(J+1))$, and the change in the vibrational energy $\Delta E_v = \Delta E - \Delta E_r$ were computed; these averages were scaled to per-collision averages using a mean collision time of 168 ps, consistent with the present experimental analysis. Using the semi-classical assumption that $\Delta E_{vib}/\omega_{bend}$ is the per-collision probability of vibrational excitation allows for the interpretation of Fig. 4d as a probability distribution of excitation across initial $J$ states. Here, we used $\omega_{bend} = 668$ cm$^{(-1)}$ to calculate example probabilities for $J = 40$ and $J = 100$ from Fig. 4d.

Vibrational mode energies were monitored along each trajectory by projecting the target collider's instantaneous momenta onto equilibrium normal modes after aligning rotational axes. To convert projected vibrational energies to quantum numbers $\epsilon_i$ that are approximately conserved before and after the collision, we averaged the vibrational mode energy $\epsilon_i$ over 250 fs at the start and end of the collision and used the usual semiclassical relationship $\epsilon_i = (v_i + 1/2)\omega_i$. Because of the strong coupling of the linear bending mode energy and rotational energy, this approach poorly describes the bending energy. Instead, we used this approach to confirm negligible excitation to the symmetric and antisymmetric stretches. Changes to the total energy were therefore attributed to changes in the rotational state and to excitation of the bending mode.

## Data availability

All data to compose the figures of this paper are available in an online repository (https://doi.org/10.5281/zenodo.7884924). All datasets generated in the current study are further available from the corresponding author upon request.

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

## Acknowledgements

This work was supported by the Office of Chemical Sciences, Geosciences, and Biosciences, Office of Basic Energy Sciences, U.S. Department of Energy. Sandia National Laboratories is a multi-mission laboratory managed and operated by National Technology and Engineering Solutions of Sandia, LLC, a wholly owned subsidiary of Honeywell International, Inc., for the U.S. Department of Energy's National Nuclear Security Administration. Argonne is a U.S. Department of Energy laboratory managed by UChicago Argonne, LLC, under Contract Number DE-AC02-06CH11357. The views expressed in the paper do not necessarily represent the views of the U.S. Department of Energy or the United States Government.

## Author contributions

T.Y.C., S.A.S., B.D.P., and C.J.K. developed the experimental apparatuses used in this work. T.Y.C. and S.A.S. performed the experiments. T.Y.C. analyzed the experimental data and prepared the paper draft. A.W.J. performed the trajectory simulations and provided the corresponding figures. C.J.K. supervised the research. All authors discussed the results and the paper.

## Competing interests

The authors declare no competing interests.
