## [Peer Review File · Nature Communications]

Direct observation of coherence transfer and rotational-to-vibrational energy exchange in optically centrifuged CO₂ superrotorsREVIEWER COMMENTS

Reviewer #1 (Remarks to the Author):

The manuscript reports on the rotational excitation of CO₂ molecules (I am guessing initially at room temperature) to superrotor states using optical centrifuge pulses. The authors were able to obtain well-resolved measurements of the rotational ladder of states and extract more accurate centrifugal distortion constants for CO₂ that are valid for a larger range of J quantum numbers in the ground vibrational state. A significant finding of this study is the observation of side bands in the Raman spectrum during field-free relaxation, which the authors attribute to the coherence transfer to the vibrational bending excitation of CO₂ as a result of molecular collisions.

I'm very much in favour of this manuscript being published in Nature Communications due to its challenging and well-executed experimental work and direct observations of the rotational-to-vibrational state coherence transfer.

However, I have a few comments.

- Firstly, the authors should consider citing several more works as previous applications of the optical centrifuge, such as, for example, molecular 3D alignment (<https://doi.org/10.1103/PhysRevLett.116.183001>), control of chiral molecules (<https://doi.org/10.1103/PhysRevLett.122.223201>, <https://doi.org/10.1103/PhysRevLett.123.243202>), and controlling the molecular rotation axis (<https://doi.org/10.1103/PhysRevResearch.3.023188>).
- Furthermore, the language in the manuscript could be improved because words are missing in some sentences and jargon, such as "odd/even rotational levels," is used.
- Most importantly, it is not entirely clear to me from reading the manuscript why the optical centrifuge cannot excite a wavepacket with some distribution over the vibrational states in the first place, which then evolves in field-free with some revivals. The authors briefly touch on this at the end of the manuscript, using the argument that rotational linewidths remain relatively constant. However, I believe this explanation deserves more discussion and expansion.

Reviewer #2 (Remarks to the Author):

The paper by Kliewer and collaborators reports on the observation of rotational-to-vibrational (R-V) energy transfers in a centrifuged CO₂ superrotor. The authors employ state-of-the-art instrumentation to observe the elusive R-V energy transfer, which is less common than the V-R phenomenon. The experimental findings are complemented and supported with theoretical simulations that further

strengthen and rationalize the conclusion. The results are sound and nicely presented and the overall investigation is compelling. The interpretation of the results as well as the manuscript are competently presented. While the results presented here might not be of interest to a broad audience, they will definitely have a high impact in more specialized areas. I believe that the novelty and fundamental interest of the current study can be a good fit for the stringent requirements of Nature Communications. In what follows, I will enumerate a number of points that the authors might want to consider to improve the already good manuscript. They are aimed at improving the clarity of the manuscript and its readability for the non-specialist readership of the journal.

1-The introduction needs to be expanded. In particular, I think the authors do a good job at introducing why their results might be relevant for other areas of research, but there are concepts that in my opinion need to be introduced in more detail for the non-specialist. For example, some expansion on superrotors would be of help as well as the effect of collisions in energy transfers. The authors provide references, but some expansion in the text would help guide the reader.

2-Also in the introduction, the authors explain that this is the first observation that superrotors can transfer rotational energy to vibrational energy. The non-specialist might want to know why this is important and therefore it needs to be put in a broader context.

3-"To our knowledge, these are the first state-resolved observations of CO₂ spinning with energies beyond the ground-state bond dissociation energy as well as coherence decay measurements of molecules rotating up to 5.5 times during a single collision" The origin of "5.5 times" is not properly introduced and why this is different compared to other studies need to be clarified.

4-In figure 1, the trace corresponding to Wu et al. is buried under the current results. I think this needs to be explained in the caption or illustrated differently for the sake of clarity.

5-Why was the pressure 380 Torr? Other experiments in the literature were performed at 50 Torr. Then in figure 2, the variation with pressure is plotted. This is confusing.

6-An expansion on D, H, and L distortions constants needs to be given as well as B, which are typical parameters in rotational spectroscopy, but they might be not so clear for the non-specialist. Also, an explanation of the fitting procedure is missing and necessary. Some details are given in the supporting information, but this needs to be either further explained in the text or/and referred to the SI.

7-Starting on page 11, the authors explore the behavior of the hot band when the centrifuge pulse is cut shorter. I find this discussion a little puzzling and hard to follow. It is not clear to me how the contribution of the R-V energy transfer is disentangled from the thermal population that could contribute to the hot band appearance.

8-The number of rotations per collision needs to be connected to the superrotor limit.

9-As it currently stands, the manuscript lacks a paragraph where the main findings of the study are summarized.

Reviewer #3 (Remarks to the Author):

The paper presents the observation of rotation-to-vibration energy transfer in CO₂ molecules, excited to extremely high rotational states (up to J=364) using an optical centrifuge.

I find the results important, since they provide insights into the very highly excited states, which are usually not populated at room temperature (the effective rotational temperature reaches 70000 K in this experiment) and therefore cannot be accessed using conventional IR or MW spectroscopies.

Thus, one can foresee applications in astrophysics as well as a deeper understanding as to what happens with rapidly rotating molecules and how the rotational angular momentum is redistributed into internal vibrations and due to molecule-molecule collisions.

The paper is well written and will be interesting to a broad community of physicists and chemists. Therefore I recommend publication in Nature Comm.

We would like to thank the reviewers for their constructive reviews of this work and for each recommending publication in Nature Communications. Below we have compiled a list to each comment, ordered by reviewer. Any changes to the manuscript are highlighted in the responses below and are labeled by page number.

Reviewer #1 (Remarks to the Author):

The manuscript reports on the rotational excitation of CO₂ molecules (I am guessing initially at room temperature) to superrotor states using optical centrifuge pulses. The authors were able to obtain well-resolved measurements of the rotational ladder of states and extract more accurate centrifugal distortion constants for CO₂ that are valid for a larger range of J quantum numbers in the ground vibrational state. A significant finding of this study is the observation of side bands in the Raman spectrum during field-free relaxation, which the authors attribute to the coherence transfer to the vibrational bending excitation of CO₂ as a result of molecular collisions.

I'm very much in favour of this manuscript being published in Nature Communications due to its challenging and well-executed experimental work and direct observations of the rotational-to-vibrational state coherence transfer.

However, I have a few comments.

- Firstly, the authors should consider citing several more works as previous applications of the optical centrifuge, such as, for example, molecular 3D alignment (<https://doi.org/10.1103/PhysRevLett.116.183001>), control of chiral molecules (<https://doi.org/10.1103/PhysRevLett.122.223201>, <https://doi.org/10.1103/PhysRevLett.123.243202>), and controlling the molecular rotation axis (<https://doi.org/10.1103/PhysRevResearch.3.023188>).

The following references have been added as examples of previous applications of the optical centrifuge:

[11] A. Milner, A. Korobenko, V. Milner, Sound emission from the gas of molecular superrotors. *Optics Express* 23(7), 8603–8608 (2015)

[16] A.A. Milner, J.A. Fordyce, I. MacPhail-Bartley, W. Wasserman, V. Milner, I. Tutunnikov, I.S. Averbukh, Controlled enantioselective orientation of chiral molecules with an optical centrifuge. *Physical Review Letters* 122(22), 223,201 (2019)

[18] E.J. Zak, A. Yachmenev, J. Kupper, Controlling rotation in the molecular frame with an optical centrifuge. *Physical Review Research* 3(2), 023,188 (2021)

Page 2: "Optical centrifuges have also been used to study molecular potentials[7], remote magnetization of gases[15], control of chiral molecules [16], three-dimensional alignment of molecules [17, 18], chemical reactivity[19], and new spectroscopic transitions[20]."

- Furthermore, the language in the manuscript could be improved because words are missing in some sentences and jargon, such as "odd/even rotational levels," is used.

The following sentences have been edited for clarity:

Page 5: “After approximately 500 ps or 3 mean collision times at 380 Torr, odd rotational energy level peaks corresponding to vibrationally excited CO₂ emerge in the pure rotational CARS spectra...”

Page 9: “The rotational CARS spectrum of CO₂ only contains even rotational energy levels at room temperature and requires high temperatures to populate bending vibrational modes for odd rotational energy levels to appear [45]. With no vibrational excitation, odd rotational energy levels lie in between the even rotational energy levels with a spacing of 1.5 cm⁻¹.”

Several typos were corrected:

Page 2: This violates traditional notions of molecular collisions as “sudden” and rotationally frozen [15, 16].

Page 6: This demonstrates that the fastest spinning CO₂ molecules observed here are highly distorted by centrifugal forces.

Page 6: This motivates the question of whether spectroscopic constants determined at lower J can extend far into the super-rotor regime.

Page 8: The constants determined by Long et al. [37] do not include L, which explains the deviation at high J.

Page 15: To explain their unusually long coherence life times, we note that so-called “motional narrowing”

- Most importantly, it is not entirely clear to me from reading the manuscript why the optical centrifuge cannot excite a wavepacket with some distribution over the vibrational states in the first place, which then evolves in field-free with some revivals. The authors briefly touch on this at the end of the manuscript, using the argument that rotational linewidths remain relatively constant. However, I believe this explanation deserves more discussion and expansion.

Here, we observed pure rotational transitions from the excited vibrational states. Ultimately, since the coherence decays are not supposed to be very sensitive to the vibrational state, any vibrationally excited CO₂ excited in the initial wavepacket should be visible at early time delays and decay together with the ground state rotational lines at approximately the same rate. We do not observe any vibrationally excited CO₂ at early time delays (see Supplementary Materials Fig. S6) and instead see these lines grow into the spectrum after several collisions. Furthermore, we observed the vibrationally excited CO₂ with reduced centrifuge bandwidths much less than the bending frequency of CO₂ (667 cm⁻¹). Additionally, recurrences are pressure-independent. Yet the arrival of vibrational hot bands is clearly correlated with pressure and enters the spectrum sooner with larger collisional frequency. Therefore, it is not likely that the observed vibrationally excited CO₂ rotational lines were directly excited by the optical centrifuge. To further clarify this point, we have amended this section as follows:

Page 14: “The vibrational state specific CO₂ rotational linewidths do not change by more than 15% [48]. Any vibrationally excited CO₂ should then be visible in the spectrum at early time delays and decay together with the vibrational ground state rotational lines. However, we do

not observe any hot bands at early time delays and instead see vibrationally excited CO₂ grow into the spectrum after several collisions occurred (see Supplementary Materials Fig. S6).”

Reviewer #2 (Remarks to the Author):

The paper by Kliewer and collaborators reports on the observation of rotational-to-vibrational (R-V) energy transfers in a centrifuged CO₂ superrotor. The authors employ state-of-the-art instrumentation to observe the elusive R-V energy transfer, which is less common than the V-R phenomenon. The experimental findings are complemented and supported with theoretical simulations that further strengthen and rationalize the conclusion. The results are sound and nicely presented and the overall investigation is compelling. The interpretation of the results as well as the manuscript are competently presented. While the results presented here might not be of interest to a broad audience, they will definitely have a high impact in more specialized areas. I believe that the novelty and fundamental interest of the current study can be a good fit for the stringent requirements of Nature Communications. In what follows, I will enumerate a number of points that the authors might want to consider to improve the already good manuscript. They are aimed at improving the clarity of the manuscript and its readability for the non-specialist readership of the journal.

1-The introduction needs to be expanded. In particular, I think the authors do a good job at introducing why their results might be relevant for other areas of research, but there are concepts that in my opinion need to be introduced in more detail for the non-specialist. For example, some expansion on superrotors would be of help as well as the effect of collisions in energy transfers. The authors provide references, but some expansion in the text would help guide the reader.

Thank you for this comment. We have expanded the introduction regarding collisions and super-rotors:

“For collisions involving super-rotors, the super-rotors can be thought of as molecular gyroscopes, where high angular momentum stabilizes the super-rotor orientation [13], promotes rotationally adiabatic collisions, and prolongs synchronized rotation [10]. In particular, entry into the rotationally adiabatic regime requires a molecule to rotate more than once during a collision. This violates traditional notions of molecular collisions as “sudden” and rotationally frozen [15, 16]. Therefore, super-rotors are typically defined by the requirement that the molecules make at least one rotation within the duration of a collision or equivalently have an adiabaticity parameter greater than π (see Supplementary Materials) [10]. The eventual thermalization of super-rotors can produce macroscopic phenomena such as vortex flows [12] as well as acoustic waves [11, 12].”

2-Also in the introduction, the authors explain that this is the first observation that superrotors can transfer rotational energy to vibrational energy. The non-specialist might want to know why this is important and therefore it needs to be put in a broader context.

The following has been added for more context regarding R-V energy transfer:

Page 3: “Quantitative understanding of R-V energy transfer could have significant impacts on modelling of hypersonic planetary entry flows and spacecraft heat shield design.”

3-“To our knowledge, these are the first state-resolved observations of CO₂ spinning with energies beyond the ground-state bond dissociation energy as well as coherence decay measurements of molecules rotating up to 5.5 times during a single collision” The origin of “5.5 times” is not properly introduced and why this is different compared to other studies need to be clarified.

This was addressed later in the paper as part of the discussion, but was not expanded upon in the introduction. The duration of one collision and rotational period was calculated as in Ref. 10 and is expanded upon in the Supplementary Materials. This has been clarified as follows:

Page 2: “In particular, entry into the rotationally adiabatic regime requires a molecule to rotate more than once during a collision. This violates traditional notions of molecular collisions as “sudden” and rotationally frozen [15, 16]. Therefore, super-rotors are typically defined by the requirement that the molecules make at least one rotation during the duration of a collision or equivalently have an adiabaticity parameter greater than π (see Supplementary Materials)[10].

Page 4: “To our knowledge, these are the first state-resolved observations of CO₂ spinning with energies beyond the ground-state bond dissociation energy as well as coherence decay measurements of molecules rotating up to 5.5 times during a single collision. This is more than three times as many rotations during a single collision as compared to past studies [10, 35]

Page 14: “As a consequence, the measured linewidths are, to our knowledge, the first to involve molecules spinning more than 5 times during a single collision, far above the super-rotor threshold of one rotation during a collision.”

4-In figure 1, the trace corresponding to Wu et al. is buried under the current results. I think this needs to be explained in the caption or illustrated differently for the sake of clarity.

This has been clarified in the caption of figure 1 as follows:

Page 5: “The constants of Wu et al. closely match those of the current work and produce an error of 0.011 cm⁻¹ at high J. Constants determined in this work produce an error of 0.0065 cm⁻¹ at high J (limited by the VIPA resolution).”

5-Why was the pressure 380 Torr? Other experiments in the literature were performed at 50 Torr. Then in figure 2, the variation with pressure is plotted. This is confusing.

Fig. 2C was used to show that the observed CO₂ hot bands were indeed collisional in nature. Due to the quadratic scaling of the CARS signal intensity with gas number density, higher pressures were preferred for resolving the vibrational hot bands. However at higher pressures, the gas collisional frequency increases. At 380 Torr, we estimate that the mean time between molecular collisions is ~168 ps. Due to the lack of ultrafast temporal resolution, experiments performed using tunable diode laser absorption

spectroscopy required pressures of 50 Torr and below to resolve dynamics on collisional time scales. The probe laser used in this study has a full-width at half maximum (FWHM) pulse width of 65 ps, which is sufficient to resolve the CARS signal below the mean time between molecular collisions at 380 Torr. At the same time, the CARS signal is maximized due to higher gas number density. Therefore, most of the measurements were performed at 380 Torr.

6-An expansion on D, H, and L distortions constants needs to be given as well as B, which are typical parameters in rotational spectroscopy, but they might be not so clear for the non-specialist. Also, an explanation of the fitting procedure is missing and necessary. Some details are given in the supporting information, but this needs to be either further explained in the text or/and referred to the SI.

The expression for the rotational energy expanded up to L has been added to the text:

Page 6: "The observed Raman shifts can be derived from following expression for the rotational energy and the S-branch selection rule of $\Delta J = 2$:

$$E_{\text{rot}}(J) = B_v J(J + 1) - D_v (J(J + 1))^2 + H_v (J(J + 1))^3 + L_v (J(J + 1))^4 \quad (1)$$

where B_v , D_v , H_v , and L_v are the vibrational level dependent rotational constant and centrifugal distortion constants, respectively. Note that sensitivity to the higher order centrifugal constants H_v and L_v requires high spectral resolution or large J . Details on the spectral fitting can be found in the Supplementary Materials."

7-Starting on page 11, the authors explore the behavior of the hot band when the centrifuge pulse is cut shorter. I find this discussion a little puzzling and hard to follow. It is not clear to me how the contribution of the R-V energy transfer is disentangled from the thermal population that could contribute to the hot band appearance.

We have clarified the caption for Fig. 4 and its corresponding discussion. Fig. 4C, D, and E represent probabilities of excitation and energy transfer for a target molecule at a known initial rotational state with a randomly sampled molecule from a 300 K thermal bath. The R-V transfer probabilities shown here are simulated with respect to the thermal bath gas, so they are not disentangled. The predicted trends matched well with the experimental data, which provides confidence that the bath gas is close to 300 K. The caption for Fig 4 will be expanded to emphasize that these simulations are produced using random sampling from a 300 K thermal bath.

For the discussion regarding the smallest centrifuge bandwidth ($J_{\text{max}} = 80$), we observed that the hot band distribution was shifted to $J < 50$ ($\Delta \nu_{\text{Raman}} < 80 \text{ cm}^{-1}$). One important aspect to note is that all shown simulation results, including probability of R-V energy transfer are with respect to the target molecule. While rotationally excited molecules lose 30 quanta of rotational energy, they also have ~18 to 20% probability of gaining vibrational energy. This was not explicitly emphasized, but it provides the basis for the argument that limiting the maximum rotational state to $J = 80$ specifically makes hot bands at $J > 50$ unlikely. It may also be confusing to the reader regarding when we are talking about the simulations or comparison of simulations with the experiment.

We have clarified this discussion as follows:

Page 11, Fig.4: “ Predicted collisional (C) rotational energy, (D) vibrational energy, and (E) angular momentum transfer with a 300 K thermal bath of CO₂. The x-axis represents the initial state of the target molecule, and its collision partner was randomly sampled from the thermal bath.”

Page 12: “The simulations show that moderate rotational excitation is efficient in driving R-V energy transfer... This explains why experimentally cutting the centrifuge pulse to $J_{\max} = 114$ does not eliminate the hot bands as these rotational states still have high probability of R-V energy transfer.”

Page 12: “As shown in Fig. 4C, the average rotational energy transfer in a 300 K thermal CO₂ bath has a maximum for CO₂ molecules with initial rotational quantum numbers around $J = 90$.”

Page 12: “Fig. 4D can be interpreted as a probability of rotation-to-vibration energy transfer in a single collision (see Supplementary Materials), with 7% probability and 25% probability for one quantum of bending excitation with initial $J = 40$ and $J = 100$...”

Page 13: “On average, molecules with initial rotational states from $J = 60$ to $J = 80$ will lose 30 quanta of rotational energy and may simultaneously gain vibrational energy. This makes observations of hot bands at $J > 50$ unlikely when $J_{\max} = 80$.”

8-The number of rotations per collision needs to be connected to the superrotor limit.

This has been amended in the expansion of the introduction as follows:

Page 2: “In particular, entry into the rotationally adiabatic regime requires a molecule to rotate more than once during a collision. This violates traditional notions of molecular collisions as “sudden” and rotationally frozen [15, 16]. Therefore, super-rotors are typically defined by the requirement that the molecules make at least one rotation during the duration of a collision or equivalently have an adiabaticity parameter greater than π (see Supplementary Materials)[10]. The eventual thermalization of super-rotors can produce macroscopic phenomena such as vortex flows [12] as well as acoustic waves [11, 12].”

Page 14: “As a consequence, the measured linewidths are, to our knowledge, the first to involve molecules spinning more than 5 times during a single collision, far above the super-rotor threshold of one rotation during a collision.”

9-As it currently stands, the manuscript lacks a paragraph where the main findings of the study are summarized.

Thank you for this comment. We have added an additional paragraph at the end that summarizes our findings:

Page 16: “In summary, R-V energy exchange and coherence transfer in optically centrifuged CO₂ was investigated using high spectral and temporal resolution coherent Raman scattering and quasi-classical trajectory simulations. From the VIPA-resolved measurements, a new set of centrifugal distortion constants was determined with less than 0.0065 cm^{-1} of error at the highest

rotational states. This resolved the disagreement between constants inferred from multi-vibrational band fits of atomic clock-stabilized frequency comb measurements in the literature. Furthermore, using ultrafast coherent Raman scattering, we have observed vibrationally excited CO₂ after three mean collision times as a result of R-V energy transfer. Trajectory simulations showed that R-V energy transfer probability peaked at moderate rotational excitation, with up to 25% probability per collision. The fastest spinning CO₂ molecules ($J>200$) were found to not have a significant role in R-V energy transfer. Lastly, the hot band transitions were found to persist unusually long, and these transitions were hypothesized to be sustained by coherence transfer. The presented measurements and simulations open the door to fully resolved pure rotational molecular structure and centrifugal distortions beyond bond-dissociation energies as well as studies of super-rotor R-V energy transfer and coherence transfer occurring on the time scale of molecular collisions.”

Reviewer #3 (Remarks to the Author):

The paper presents the observation of rotation-to-vibration energy transfer in CO₂ molecules, excited to extremely high rotational states (up to $J=364$) using an optical centrifuge.

I find the results important, since they provide insights into the very highly excited states, which are usually not populated at room temperature (the effective rotational temperature reaches 70000 K in this experiment) and therefore cannot be accessed using conventional IR or MW spectroscopies.

Thus, one can foresee applications in astrophysics as well as a deeper understanding as to what happens with rapidly rotating molecules and how the rotational angular momentum is redistributed into internal vibrations and due to molecule-molecule collisions.

The paper is well written and will be interesting to a broad community of physicists and chemists. Therefore I recommend publication in Nature Comm.

Thank you for these comments. The positive feedback is much appreciated by the authors.

Other changes:

In the acknowledgements, the following was added:

“Argonne is a U.S. Department of Energy laboratory managed by UChicago Argonne, LLC, under Contract Number DE-AC02-06CH11357”

Author contributions have also been added:

“TYC, SAS, BDP, and CJK developed the experimental apparatuses used in this work. TYC and SAS performed the experiments. TYC analyzed the experimental data and prepared the manuscript draft. AWJ performed the trajectory simulations and provided the corresponding figures. CJK supervised the research. All authors discussed the results and the manuscript.”

REVIEWERS' COMMENTS

Reviewer #1 (Remarks to the Author):

I appreciate the authors' clarifications and amendments to the manuscript. I hereby support the publication of the manuscript in its current form in Nature Communications.

Reviewer #2 (Remarks to the Author):

The authors have fully addressed my comments and those of other reviewers. In my opinion, the authors have considerably improved the manuscript. I recommend the publication of this very well-executed and relevant study.